# The gut microbiota composition is linked to subsequent occurrence of ventilator-associated pneumonia in critically ill patients

Arthur Orieux,[1] Raphaël Enaud,[2,3] Sébastien Imbert,[2,4] Philippe Boyer,[1] Erwan Begot,[1] Adrian Camino,[2] Alexandre Boyer,[1,2] Patrick Berger,[2] Didier Gruson,[1,2] Laurence Delhaes,[2,4] Renaud Prevel[1,2]

**ABSTRACT**   Ventilator-associated pneumonia (VAP) is the most frequent nosocomial infection in critically ill-ventilated patients. Oropharyngeal and lung microbiota have been demonstrated to be associated with VAP occurrence, but the involvement of gut microbiota has not been investigated so far. Therefore, the aim of this study is to compare the composition of the gut microbiota between patients who subsequently develop VAP and those who do not. A rectal swab was performed at admission of every consecutive patient into the intensive care unit (ICU) from October 2019 to March 2020. After DNA extraction, V3-V4 and internal transcribed spacer 2 regions deep-sequencing was performed on MiSeq sequencer (Illumina) and data were analyzed using Divisive Amplicon Denoising Algorithm 2 (DADA2) pipeline. Among 255 patients screened, 42 (16%) patients with invasive mechanical ventilation for more than 48 h were included, 18 (43%) with definite VAP and 24 without (57%). Patients who later developed VAP had similar gut bacteriobiota and mycobiota α-diversities compared to those who did not develop VAP. However, gut mycobiota was dissimilar (β-diversity) between these two groups. The presence of *Megasphaera massiliensis* was associated with the absence of VAP occurrence, whereas the presence of the fungal genus *Alternaria* sp. was associated with the occurrence of VAP. The composition of the gut microbiota, but not α-diversity, differs between critically ill patients who subsequently develop VAP and those who do not. This study encourages large multicenter cohort studies investigating the role of gut-lung axis and oropharyngeal colonization in the development of VAP in ICU patients. Trial registration number: NCT04131569, date of registration: 18 October 2019.

**IMPORTANCE**   The composition of the gut microbiota, but not α-diversity, differs between critically ill patients who subsequently develop ventilator-associated pneumonia (VAP) and those who do not. Investigating gut microbiota composition could help to tailor probiotics to provide protection against VAP.

**KEYWORDS**   ventilator-associated pneumonia, microbiota, mycobiota, intensive care unit

Lower respiratory tract infections are the leading causes of nosocomial infections in intensive care unit (ICU) (1, 2) and can be distinguished by whether it occurs in the presence of invasive mechanical ventilation or not. Ventilator-associated pneumonia (VAP) is defined as pneumonia occurring at least 48 h after invasive mechanical ventilation (3). VAP affects 20%–40% of patients receiving invasive mechanical ventilation but more than 70% of the severe patients with longer invasive mechanical ventilation. Indeed, the incidence of VAP is positively correlated with the duration of invasive mechanical ventilation (about 10–25 VAP for 1,000 days of mechanical

Address correspondence to Renaud Prevel, renaud.prevel@hotmail.fr.

The authors declare no conflict of interest.

See the funding table on p. 11.

ventilation) (3–5). VAP occurrence is associated with severe morbidity leading to increased duration of invasive mechanical ventilation, ICU stay, and hospitalization lengths (6, 7).

Despite recent advances in understanding pathophysiological mechanisms responsible for VAP, very few have been in preventive measures. Microbiota has been proven to be involved in numerous chronic respiratory diseases and in acute respiratory infections such as influenza or bacterial pneumonia (8). This role can be mediated via microbial direct interactions alleviating or enhancing host colonization resistance and the emergence of pathogens (9, 10). Microbiota also interacts with the host modulating the local and systemic immune system (11, 12). Lung microbiota was logically the more extensively investigated compartment as VAP develops within the lungs (13). Reduced lung bacteriobiota α-diversity was associated with increased pathogenic bacterial presence and increased lung inflammation (14). Consistent with the concept of transcolonization, the composition of oropharyngeal and gut microbiota could influence lung microbiota composition (8, 15). The composition of gut microbiota can also enhance host resistance to lung bacterial infection *via* lung immune system modulation (12, 16). Despite this pre-clinical data, the role of gut microbiota in VAP occurrence has never been investigated. We thus aim to compare the gut bacterial and fungal microbiota (respectively, bacteriobiota and mycobiota) composition between patients who subsequently develop VAP and those who do not.

## MATERIALS AND METHODS

### Patients' inclusion and data collection

This study is an ancillary study from Microbe study (NCT04131569) which prospectively included every consecutive patient older than 18 years of age admitted to the medical ICU at Bordeaux University Hospital from October 2019 to March 2020 (stopped at COVID-19 wave occurrence). Rectal swabs (Transport Swab VWR, Copan) used for fecal extended-spectrum beta-lactamase (ESBL)-E carriage screening at admission, before administration of antimicrobial therapy, were collected and frozen at −80°C. Only samples sufficiently loaded, as assessed by visual control, with fecal materials from patients with invasive mechanical ventilation >48 h were included as poorly loaded swabs are dominated by peri-rectal skin bacteria and sequencing contaminants (17).

Data were prospectively recorded by physicians in charge of the patient by questioning the patients, patients' family, and patients' general practitioners. Electronic worksheets were completed by two medical intensive care residents. Comorbidities were defined as follows: chronic obstructive pulmonary disease and asthma were defined according to prior lung function testing. Chronic heart failure was defined according to prior transthoracic echocardiography and chronic coronary disease based on a prior stress test or percutaneous coronary intervention. Other comorbidities included history of chronic kidney disease, immunosuppression (drugs, hematological disease, blood marrow transplantation, solid organ transplantation, plasma exchanges indicated by autoimmune disorders, human immunodeficiency virus infection), and simplified acute physiology score II (SAPSII). Acute respiratory distress syndrome was defined according to Berlin's criteria (18), septic shock according to SEPSIS-III definition (19), and acute kidney injury according to kidney disease improving global outcomes (KDIGO) guidelines (20).

### Definition of ventilator-associated pneumonia

The patients' charts were reviewed by two independent ICU physicians who allocated patients to the following groups: "definite VAP" or "absence of definite VAP". In the case of disagreement, a third senior investigator was consulted to achieve consensus (needed for one patient only). "Definite VAP" was diagnosed when ventilated patients presented clinical and radiological signs of VAP after at least 2 days of invasive mechanical

ventilation with a clinical pulmonary infection score > 6 (21) and ≥$10^5$ bacteria/mL on endotracheal aspirates or ≥$10^4$ bacteria/mL in bronchoscopic broncho-alveolar lavage fluid.

## DNA extraction, library preparation, and statistical analyses

DNA extraction was performed using QIAamp PowerFaecal Pro DNA kit (QIAgen, Valencia, CA, USA). A first step of mechanical lysis (2 cycles of 30 s at 7,000 rpm on Precellys evolution) was added to the chemical lysis of the kit as previously described (22). The gut microbiota and mycobiota composition of samples was assessed, respectively, by using the V3-V4 regions of the bacterial 16S rRNA encoding gene and the internal transcribed spacer 2 (ITS2) region of the fungal rDNA. The respective primers used to amplify these loci were as follows: 16S-forward, TACGGRAGGCAGCAG; 16S-reverse, CTACCNGGGTATCTAAT; ITS2-forward, GTGARTCATCGAATCTTT; and ITS2-reverse, GATATGCTTAAGTTCAGCGGGT. Sequencing (2 × 250 bp paired-end) was performed on MiSeq sequencer (Illumina, San Diego, CA, USA) at the Plateforme Génome Transcriptome de Bordeaux (PGTB) platform (INRAe, University of Bordeaux, Cestas, France).

The bacterial and fungal reads were demultiplexed; 16S and ITS2 primers were removed using CutAdapt, with no mismatch allowed within the primer sequences. All samples were processed through the DADA2 pipeline in R (version 4.0.3) for quality filtering and trimming, dereplication, and merging of paired-ends reads (23, 24). According to a recent evaluation (25), only forward sequences were analyzed with DADA2, and no filter other than the removal of low-quality and chimeric sequences was applied for characterizing the fungal community. We used mock communities (compositions in the Supplemental Materials) and negative controls (three from the DNA extraction step with unloaded swabs and three from the PCR amplification step) to ensure the sequencing quality. Two distinct amplicon sequence variant (ASV) tables were constructed, and taxonomy was assigned from the Silva database (release 138) for bacterial ASVs and the Unite database (release 8.2) for fungal ASVs. Comparison of β-diversity among negative control, mock community, and samples is available in Fig. S1 and S2 for bacteriobiota and mycobiota, respectively. The final average read counts (amplicons after quality filtration but before assignment to obtain ASVs) per sample were 41,620 (standard deviation ±11,751) for 882 bacterial ASVs and 4,694 (standard deviation ±6,426) for 337 fungal ASVs. The 16S rRNA gene and ITS2 sequences have been submitted to the European Nucleotide Archive (accession no. ERP134949).

For microbiota and mycobiota analysis, α-diversity metrics (Richness, Simpson, and Shannon indices) were generated by using the phyloseq R package. For cross-sectional analyses, at a specific time, significant differences in phyla abundance and in α-diversity were determined using the Mann-Whitney Wilcoxon rank-sum test. Between sample β-diversity, differences (measured using Bray-Curtis dissimilarity) were tested using a permutational multivariate analysis of variance (ANOVA) (PERMANOVA) from vegan R package with 10,000 permutations, while accounting for individual identity as a covariate. Linear discriminant analysis (LDA) effect size (LefSe) analysis was performed from microbiomeMarker package. Statistical analysis was performed with the R studio program (version 1.3.1056 for Windows); correction for multiple testing was performed using the Benjamini-Hochberg false discovery rate (FDR) procedure, a $P$ value or FDR adjusted $P$ value equal to or less than 0.05 was considered statistically significant.

## Statistical analysis

No statistical sample size calculation was performed *a priori*, and sample size was equal to the number of patients admitted to ICU during the study period.

Quantitative variables are presented as median and interquartile range (IQR), and compared by using Mann-Whitney Wilcoxon rank-sum test. Categorical variables are expressed as number of patients (percentage) and compared by mean of the $\chi^2$ or Fisher tests. All statistical tests were two-tailed and statistical significance was defined as $P <$

0.05. Statistical analyses were assessed by the R studio program (version 3.6.1056 for Windows).

## RESULTS

### Flow chart

Among the 255 patients admitted to our ICU, 83 (33%) required invasive mechanical ventilation for 48 h or more. Forty-two patients were included in further analysis, 18 (43%) with definite VAP, and 24 (57%) without (Fig. 1). None of the patients received prebiotics, probiotics, or selective digestive decontamination.

### Patients' characteristics

Patients with or without VAP had similar characteristics, severity, and causes of admission (Table 1). The proportion of patients who received long-term pomp proton inhibitors, long-term metformin, or antimicrobial therapy within the last 3 months of admission was not different between these two groups ($P = 1.00$, $P = 0.21$, and $P = 0.75$, respectively). Notably, patients who received antimicrobial therapy within the last 3 months showed no difference in the gut bacteriobiota or mycobiota α-diversities but showed dissimilar gut microbiota composition than those who did not receive any antimicrobial therapy (Fig. S3 and S4, respectively).

The microbiological results of routine culture samples are presented in Table S1. The more frequently isolated bacterial species were *Citrobacter koseri*, *Klebsiella pneumoniae*, *Klebsiella variicola*, and *Escherichia coli*. Of note, the causative VAP bacteria were present in the gut microbiota in 9/18 patients (50%) who will develop subsequent VAP with a median proportion of reads related to the causative bacteria compared to the number of reads after filtration for the corresponding sample of 0.66%, IQR (0.39–0.95). The species involved were *K. pneumoniae* ($n = 3$), *E. coli* ($n = 2$), *K. variicola* ($n = 1$), *Acinetobacter baumannii* ($n = 1$), *Enterococcus faecalis* ($n = 1$), and *Enterococcus faecium* ($n = 1$). The median time between orotracheal intubation (OTI) and VAP occurrence was 6 days, IQR (3–8) with a trend for a shorter time in patients with the causative bacteria detected in the gut microbiota at admission compared to the others (median time 4 days, IQR [3–6] vs 8 days, IQR [4–17], $P = 0.09$).

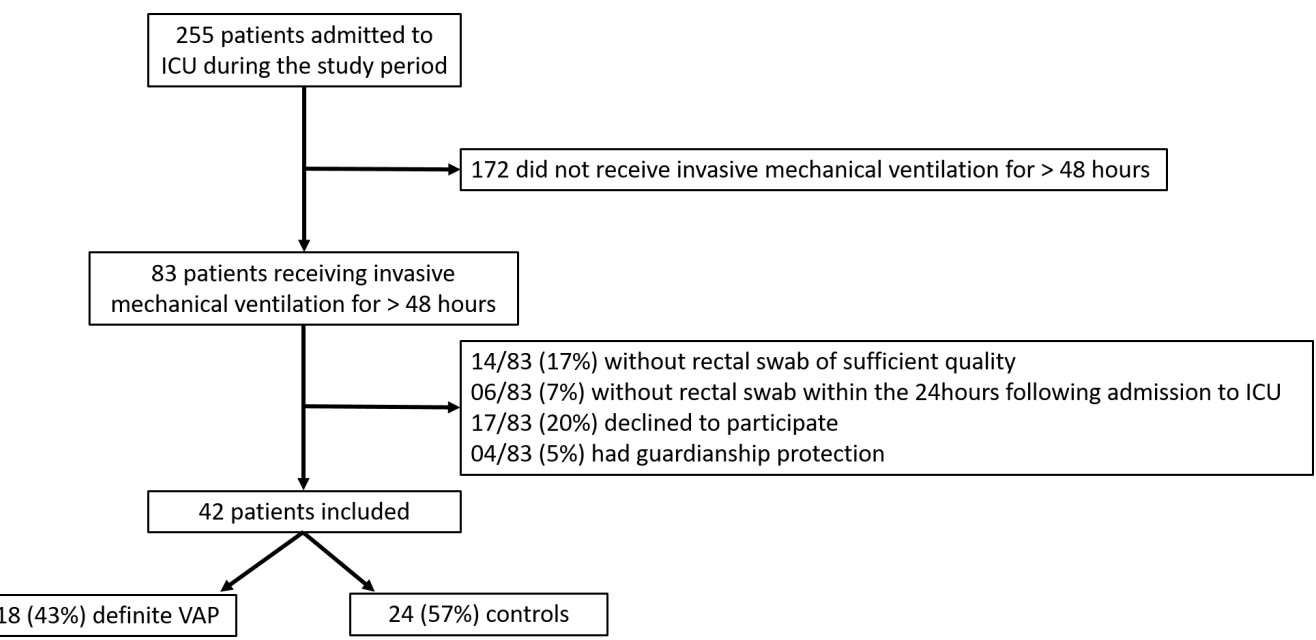

**FIG 1** Flow-chart of the patients screened and included in the study.

**TABLE 1.** Patients' characteristics and comparison between those who subsequently develop ventilator-associated pneumonia and those who do not[a]

| | Total (n = 42) | VAP (n = 18) | Controls (n = 24) | P value |
|---|---|---|---|---|
| **Patients' characteristics at admission to ICU** | | | | |
| Age | 67 (61–77) | 64 (59–74) | 70 (65–78) | 0.22 |
| Sex (male) | 28 (67%) | 14 (78%) | 14 (58%) | 0.32 |
| SAPSII | 75 (65–85) | 78 (57–83) | 75 (67–85) | 0.52 |
| Chronic pulmonary disease | 11 (26%) | 6 (33%) | 5 (21%) | 0.48 |
| COPD | 9 (21%) | 4 (22%) | 5 (21%) | 1.00 |
| Asthma | 2 (05%) | 2 (11%) | 0 (0%) | 0.18 |
| Chronic heart failure | 11 (26%) | 6 (33%) | 5 (21%) | 0.48 |
| Chronic coronary disease | 10 (24%) | 4 (22%) | 6 (25%) | 1.00 |
| Chronic kidney disease | 7 (17%) | 4 (22%) | 3 (13%) | 0.44 |
| Immunosuppression | 8 (19%) | 4 (22%) | 4 (17%) | 0.71 |
| Active solid cancer | 5 (12%) | 3 (17%) | 2 (08%) | 0.64 |
| **Causes of admission** | | | | |
| Septic shock | 13 (31%) | 4 (22%) | 9 (38%) | 0.37 |
| Coma | 13 (31%) | 7 (39%) | 6 (25%) | 0.51 |
| Acute respiratory failure | 9 (21%) | 3 (17%) | 6 (25%) | 0.61 |
| Cardiac arrest | 4 (10%) | 3 (17%) | 1 (04%) | 0.53 |
| Others | 3 (07%) | 1 (06%) | 2 (08%) | 1.00 |
| **Medications** | | | | |
| Antimicrobial treatment within the past 3 mo | 15 (36%) | 7 (39%) | 8 (33%) | 0.75 |
| Amoxicillin | | 2 | 4 | – |
| Amoxicillin + clavulanic acid | | 2 | 2 | – |
| Tazocilline + amikacin | | 0 | 2 | – |
| Tazocilline + ofloxacin | | 1 | 0 | – |
| Tazocilline then meropenem | | 1 | 0 | – |
| Third CG | | 1 | 0 | – |
| Quinolones | | 1 | 0 | – |
| Metformin | 7 (17%) | 1 (06%) | 6 (25%) | 0.21 |
| Proton pump inhibitor | 8 (19%) | 3 (17%) | 5 (21%) | 1.00 |
| **Care in ICU** | | | | |
| Septic shock | 28 (67%) | 12 (67%) | 16 (67%) | 1.00 |
| ARDS | 9 (21%) | 5 (28%) | 4 (17%) | 0.46 |
| Acute kidney injury | 29 (69%) | 11 (61%) | 18 (75%) | 0.50 |
| **Treatment** | | | | |
| Renal replacement therapy | 8 (19%) | 4 (22%) | 4 (17%) | 0.71 |
| Enteral nutrition | 31 (74%) | 12 (67%) | 19 (79%) | 0.48 |
| Enteral + parenteral nutrition | 8 (19%) | 5 (28%) | 3 (13%) | 0.26 |
| Digestive surgery | 8 (19%) | 4 (22%) | 4 (17%) | 0.71 |
| **Outcome** | | | | |
| Day 28 mortality | 12 (29%) | 7 (39%) | 5 (21%) | 0.30 |

[a]Results are presented as proportion for categorical variables and median (interquartile range) for continuous variables. Categorical data are presented as number and percentage. Antimicrobial treatment: treatment received within the past 3 months; ARDS: acute respiratory distress syndrome; and third CG: third generation cephalosporin. $P$ values are for comparison between patients with VAP and controls. Threshold for statistical significance: $P = 0.05$.

## Description of gut bacteriobiota and mycobiota

The gut microbiota of both patients who subsequently develop VAP or not was predominantly composed of bacteria from Firmicutes, Bacteroidota, and Proteobacteria phyla with Bacteroidota being more abundant in the group of patients who develop VAP ($P = 0.02$) (Fig. 2A). Fungi from Ascomycota and Basidiomycota were predominant in gut mycobiota of both patients who subsequently develop VAP or not, with no statistical difference (Fig. 2B).

## Patients who subsequently develop VAP have similar α-diversity of gut bacteriobiota and mycobiota than those who did not

The gut bacteriobiota α-diversity was similar between patients who subsequently develop VAP and those who do not, regarding Shannon and Simpson indices, and evenness ($P = 0.48$, $P = 0.41$, and $P = 0.47$, respectively) (Fig. 3A through C) as was the gut mycobiota α-diversity ($P = 0.06$, $P = 0.64$, and $P = 0.79$, respectively) (Fig. 4A through C).

## Gut mycobiota, but not bacteriobiota, is dissimilar between patients who subsequently develop VAP and those who do not

The gut mycobiota β-diversity allows to discriminate between patients who subsequently develop VAP and those who do not (PERMANOVA, $P = 0.05$) but not gut bacteriobiota (PERMANOVA, $P = 0.56$) (Fig. 3D and 4D).

## Identification of bacterial and fungal species significantly associated with subsequent VAP occurrence

Using LDA analysis to detect significant microbial community enrichment, we identified *Megasphaera massiliensis* as being associated with the absence of VAP occurrence (LDA score > 3) (Fig. 5A). This association was confirmed by analysis of compositions of microbiomes with bias correction (ANCOM-BC) analysis ($P < 0.001$). On the contrary, several bacterial species (*Bifidobacterium breve*, *Peptostreptococcus anaerobius*, and *Enterococcus avium*) were associated with the occurrence of subsequent VAP, but this association was not confirmed by ANCOM-BC analysis. This association was driven by the fact that *M. massiliensis* was not observed in any gut microbiota of patients who will develop subsequent VAP (Fig. S5A). Regarding fungi, the fungal genus (*Alternaria* sp.) and fungal species (*Saccharomyces kudriavzevii*) were associated with subsequent VAP occurrence (Fig. 5B), but this association was only confirmed for *Alternaria* sp. by ANCOM-BC analysis ($P < 0.0001$). *Alternaria* sp. was only found in the gut microbiota of patients who will develop subsequent VAP (Fig. S5B). The results of ANCOM-BC analysis are provided in Tables S2 and S3 for bacteriobiota and mycobiota analysis, respectively.

## DISCUSSION

To the best of our knowledge, this study is the first to investigate the link between gut microbiota composition and VAP occurrence in critically ill patients. Notably, we identified that patients who subsequently develop VAP have similar gut bacteriobiota and mycobiota α-diversities than those who do not but that gut mycobiota compositions were dissimilar between these two groups. Using LDA analysis, this approach could

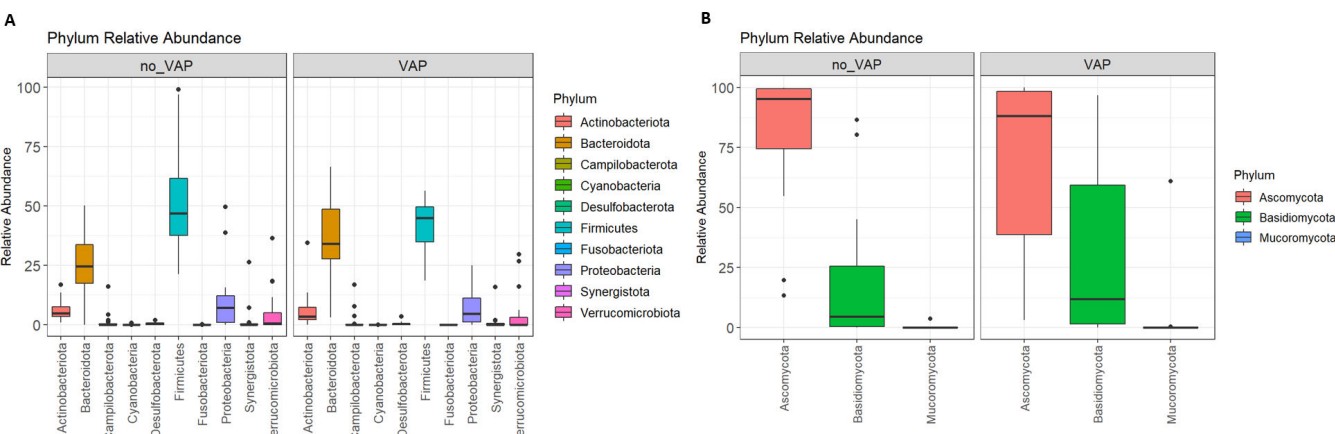

**FIG 2** Comparison of gut microbiota phylum relative abundance between patients who subsequently develop VAP and those who do not. VAP: patients who subsequently develop VAP; no_VAP: patients who do not (Wilcoxon rank-sum test); and threshold for statistical significance: $P = 0.05$.

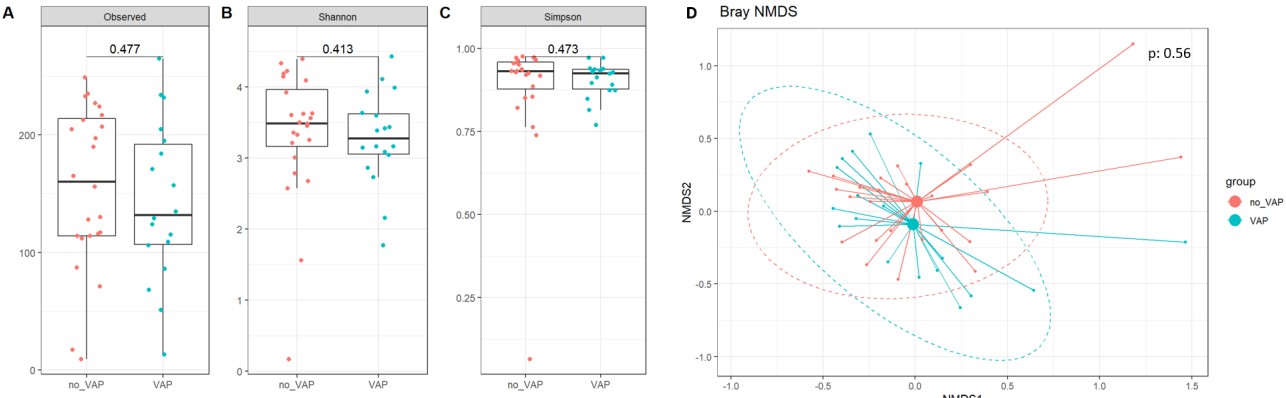

**FIG 3** Comparison of gut bacteriobiota between patients who subsequently develop VAP and those who do not. (A) Boxplot of estimated α-diversity by Shannon index. (B) Boxplot of estimated α-diversity by Simpson index. (C) Boxplot of estimated α-diversity by evenness. (D) Non-metric Bray-Curtis analysis of β-diversity. Larger filled circles indicate group centroids. Ellipses indicate the 95% confidence interval around the centroid in non-dimensional space. Threshold for statistical significance: $P = 0.05$.

identify potential probiotic candidates for VAP prevention, i.e., *M. massiliensis*, whereas several bacteria and fungi were associated with VAP occurrence.

Our population is representative of critically ill patients receiving orotracheal intubation, but the rate of VAP occurrence in this study is high (43%). Nevertheless, it remains consistent with the existing literature in patients receiving invasive mechanical ventilation for at least 2 days (1, 2).

Only two studies investigated the potential role of mycobiota in VAP development. One did not find any fungal specie of oropharyngeal microbiota to be associated with VAP (26). The other identified *Agaricomycetes* and an unclassified *Ascomycota* only in the VAP cohort (27). Even if it can be disturbing to imagine a role of gut microbiota in the development of a long-distance, other-kingdom infection such as bacterial VAP, the role of gut mycobiota in VAP should not be underestimated as demonstrated by the dissimilarity in gut mycobiota composition between patients who develop VAP or not observed in this study. In fact, direct inter-kingdom interactions occur within the microbiota (8) and could modulate the dynamics of the bacteriobiota and/or the expansion of any pathogen. For instance, *E. coli* and *Candida albicans* have been demonstrated to co-evolve within biofilms conferring increased ofloxacine tolerance to

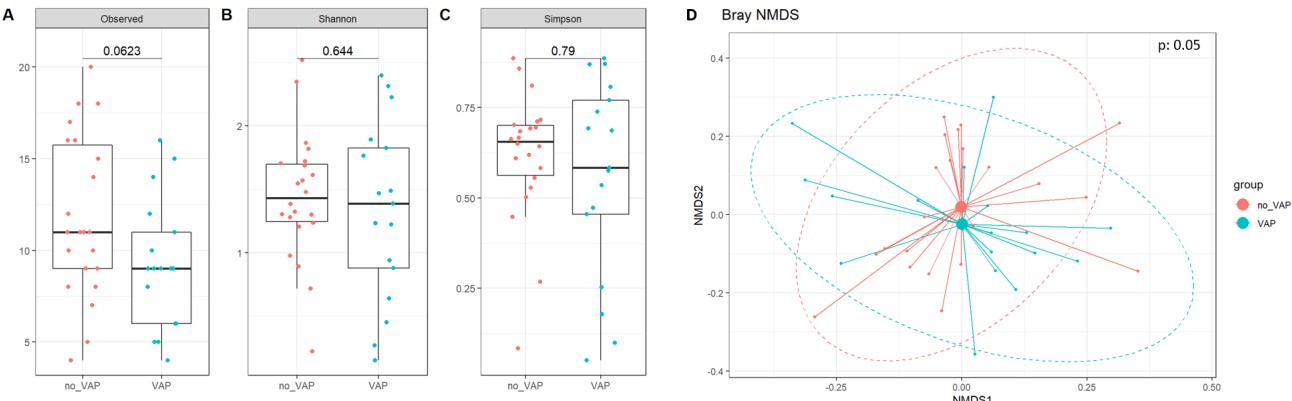

**FIG 4** Comparison of gut mycobiota between patients who subsequently develop VAP and those who do not. (A) Boxplot of estimated α-diversity by Shannon index. (B) Boxplot of estimated α-diversity by Simpson index. (C) Boxplot of estimated α-diversity by evenness. (D) Metric Bray-Curtis analysis of β-diversity. Larger filled circles indicate group centroids. Ellipses indicate the 95% confidence interval around the centroid in non-dimensional space. Threshold for statistical significance: $P = 0.05$.

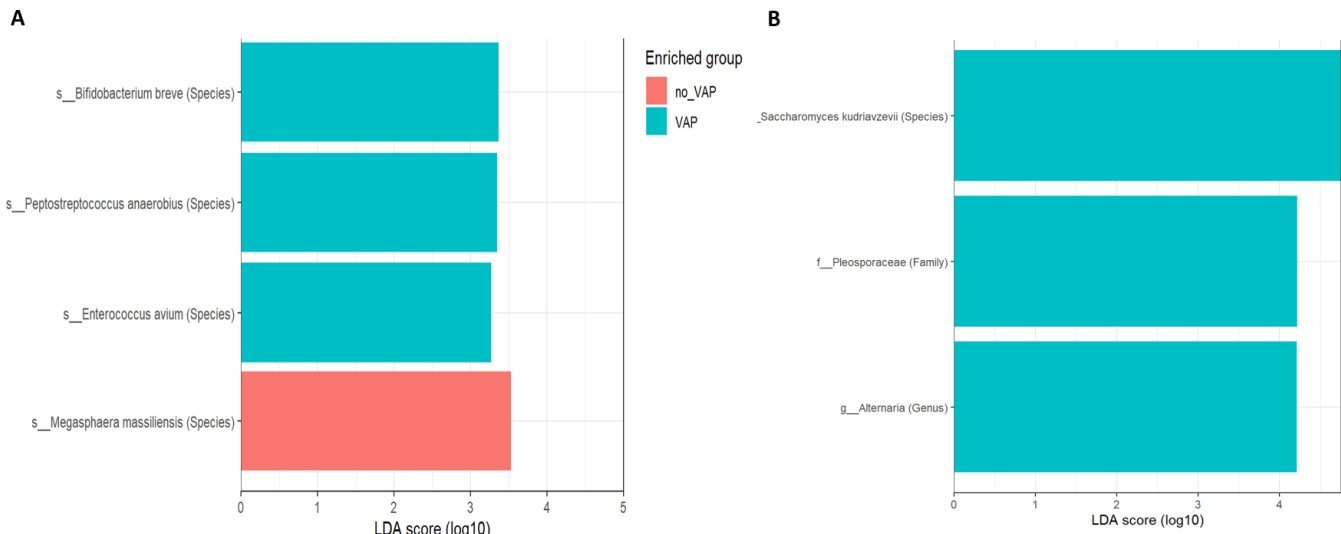

**FIG 5** Identification of bacterial and fungal species associated with subsequent occurrence of ventilator-associated pneumonia. LEfSe analysis with LDA of (A) gut bacteriobiota and (B) gut mycobiota. Threshold for statistical significance: LDA > log3.

*E. coli* (28). Besides, commensal fungi are major drivers of gut barrier integrity (29) and could help to limit bacterial translocation from the gut. Moreover, commensal fungi also have an indirect impact through the modulation of local and systemic immunity (30) enhancing the plausible relevant impact of gut mycobiota in subsequent development of VAP. This point has recently been highlighted by the enhanced granulopoiesis after rewilding laboratory mice *via* fungal colonizing, conferring long-term protection from bacterial infection to these mice (31). This study confirms a previous one which demonstrated that commensal fungi recapitulate the protective benefits of intestinal bacteria (30). Interestingly, a link between decreased gut microbiota diversity and increased rate of nosocomial infections via alteration of neutrophils functions has recently been suggested but investigation of mycobiota was lacking (32). Consequently, the fungal kingdom of microbiota should be investigated in future studies.

Thanks to the advances of next-generation sequencing, insights into the role of both oropharyngeal and lung bacteriobiotas in the occurrence of VAP have recently been made. A first study in 2012 found significant differences between the lung microbiota of patients and controls with higher abundances of bacteria belonging to *Bacilli*, *Gammaproteobacteria* in VAP patients and higher abundances of *Bacteroidia* and *Clostridia* in controls (27). In another study, lung bacteriobiota α-diversity was also decreased in patients with *Pseudomonas aeruginosa* VAP compared to controls (33). Longitudinal cohort study confirmed that the duration of mechanical ventilation, more than antimicrobial treatment, is a major determinant of the decrease in lung microbiota α-diversity in intubated patients (34). More recent studies confirmed that reduced lung bacteriobiota α-diversity was associated with increased pathogenic bacterial presence and increased lung inflammation in non-COVID-19 (14) and also in COVID-19 patients (35). The changes in lung microbiota β-diversities between patients who developed VAP and those who did not were significantly different (36) but discrepant results were obtained in the specific population of traumatic brain injury ventilated patients, with patients who developed VAP during ICU having different structures of broncho-alveolar lavage (BAL) microbiota either at admission and at 7 days post-ICU admission (37). As lungs and oropharynx are connected by the endotracheal tube (ETT) in patients receiving invasive mechanical ventilation, some studies aimed to link lung and oropharyngeal microbiota compositions. A microbial shift has been demonstrated in oropharyngeal microbiota of intubated patients with the acquisition of potential respiratory pathogens such as *Staphylococcus aureus*, *Streptococcus pseudopneumoniae*,

and *Escherichia coli* which was reversible after extubation (38). As the microbiota of dental plaque, ETT, and lungs are quite similar (39), the role of oropharyngeal microbiota in VAP occurrence has been further investigated. The first study did not demonstrate any significantly lower oropharyngeal microbiota α-diversity in patients with VAP but probably due to a limited sample size (40). Nevertheless, detection of *Enterobacteriaceae* in oropharyngeal microbiota only occurred in patients who subsequently develop *Enterobacteriaceae* VAP (40) and this colonization happened early in the course of invasive ventilation. Finally, 18 "definite" VAP were matched to 36 controls and suggested that a low relative abundance of Bacilli in the oropharyngeal microbiota at the time of intubation is associated with subsequent VAP occurrence (41). Lung and oropharyngeal microbiota compositions were statistically different and highly heterogeneous between individuals but exhibited a similar trend of changes over time (41). These data confirm the findings from the 1970s (42, 43) which demonstrated a continuum of lung colonization from the oropharyngeal cavity. However, if extended tooth brushing prevents oropharyngeal dysbiosis in patients receiving invasive mechanical ventilation, it was not effective in reducing the risk of nosocomial infections, including pneumonia (44).

Isolation of Gram-negative bacteria is increased in the oropharynx within a few hours after orotracheal intubation (45, 46). This could be explained by the fact that the digestive fluids are present in the lungs of intubated patients (47–49) and migration of radiolabeled elements from the stomach to the lungs has been evidenced (50). Numerous factors could explain this facilitated transcolonization in critically ill intubated patients as they experience both systemic and local perturbations. Regional perturbations that can increase digestive fluid reflux from the gut to the oropharynx include the posture in a supine or prone position, enteral nutrition, thoracic and abdominal pressure regimen due to pressure-positive ventilation, the presence of a gastric tube, treatments lowering esophageal sphincter tone, or corticosteroids use (15). Decreased digestive motility and an increase in gastric pH also favor the intra-gastric proliferation of *Enterobacteriaceae*; facilitating its dissemination to the oropharynx (47). All these endogenous factors and clinical interventions can deeply modify the gut microbiota composition as previously reviewed (51). Besides this transcolonization part, the gut microbiota also exerts a major influence on both systemic immunity and local lung immune response to infections (12, 16, 52). In fact, in a murine model of *K. pneumoniae* lung infection, the depletion of the gut microbiota decreased the stimulation of innate immunity by nucleotide oligomerization domain (NOD)-like receptors in the gut which impaired the reactive oxygen species-mediated killing of bacteria by alveolar macrophages (16). Peptidoglycan from the gut translocates to neutrophils in the bone marrow enhancing the neutrophils' ability to clear *Streptococcus pneumoniae* from the lungs (12). Because of the more extensive data regarding oropharyngeal and lung microbiota but the absence of data regarding gut microbiota, we decided to explore, for the first time, the role of gut microbiota in VAP occurrence.

Nevertheless, the lack of concomitant lung and oropharyngeal samples is the first limitation of our study, as lung microbiota is dependent on gut and oropharyngeal microbiota as described in the "gut-lung axis" and "transcolonization" concepts explained above. It would be relevant to investigate the interplay between the microbiota of these three compartments and its impact on the development of VAP longitudinally. Another limitation is the lack of longitudinal samples. Although the treatments received in ICU can dramatically alter the gut microbiota composition, their dynamics within the first days of invasive mechanical ventilation were not assessed (53). In addition, some patients could have received treatments before admission to ICU which could impact lung microbiota composition. Long-term pump proton inhibitors and metformin treatments or recent antimicrobial therapy are known to potentially impact the gut microbiota (54, 55), but the proportion of patients receiving those treatments was equivalent between the two groups, limiting this bias. Even if it was not the research question of this study, adjustment on all these variables would require larger cohorts including hundreds of patients. This absence of difference is consistent

with the resiliency in α-diversity between patients who received antimicrobial therapy within the past 3 months and those who had not. This absence of difference is not consistent with the resiliency of gut microbiota diversity within weeks after antibiotics administration already observed, even if this resiliency is impaired after several antimicrobial courses (56). Rectal swabs used for this study were performed before the administration of any antimicrobial therapy within the ICU. Finally, our study provides correlation but not causality demonstration. To go beyond the association links provided in this study, animal models are needed to assess the ability of identified candidate probiotics to prevent VAP occurrence and *in vitro* studies are needed to identify the underlying mechanisms if a causal link is demonstrated. These animal models and *in vitro* studies are important, especially in view of the highly discordant studies investigating the efficacy of probiotics to prevent VAP occurrence (57). Importantly, probiotics administration could even be deleterious in some subset of patients (58). A major pitfall in the interpretation of these studies is the criteria used to select probiotic candidates. Notably, differential abundance testing methods often provide different results and it is now recommended to confront at least two differential abundance methods to help ensure robust biological interpretations (59). Using this multiple tests' method, the fungal genus *Alternaria* sp. was found to be associated with the occurrence of subsequent VAP. *Alternaria* sp. is known to increase lung inflammation in murine models of asthma *via* tuft cell-produced cysteinyl leukotrienes and IL-25 (60, 61). On the contrary, the bacterial specie *M. massiliensis* is associated with the absence of VAP occurrence which is on high interest due to the ability of *M. massiliensis* to produce valerate, butyrate, and pentanoate, decreasing the production of IL-6 *in vitro*, having HDAC inhibitory function and modulating both CD4 and CD8 T cells functions (62–64). In fact, concerns exist about the translocation of probiotics given to critically ill patients with increased gut permeability (65) and it is important to note that the ability of the micro-organisms used in probiotics to colonize their environment is not the main factor of microbiota modulation. In fact, their ability to share genes and metabolites with other micro-organisms within the microbiota and to interact with host epithelial and immune cells are thought to be key mechanisms (66). As a result, VAP prevention could be mediated *via* microbial metabolites and/or products, as butyrate produced by *M. massiliensis*, rather than by the micro-organisms themselves.

## Conclusions

The composition of the gut mycobiota is dissimilar between patients who subsequently develop VAP and those who will not. *M. massiliensis* is associated with the absence of VAP occurrence and could be an interesting probiotic candidate for VAP prevention. This study is encouraging for future large multicenter cohort studies investigating the role of gut-lung axis and oropharyngeal colonization in the development of VAP in ICU patients.

### ACKNOWLEDGMENTS

We would like to thank Erwan Guichoux and Marie Massot for technical assistance. We are grateful to every ICU health worker who cared for patients and helped in samples collection. We thank Shradha Wali for English editing.

This work was funded by a grant from "Fédération Girondine de Lutte contre les Maladies Respiratoires." R.P. received a personal salary grant from CHU de Bordeaux (MD/PhD program).

R.P., P.B., L.D., and D.G. contributed to the conception and design of the study. A.O., P.B., A.B., and R.P. contributed to the acquisition of data. R.P. and A.C. performed DNA extraction. S.I. and R.E. performed bioinformatics and statistical analysis. Each author drafted or provided critical revision of the article and provided final approval of the version submitted for publication.

On behalf of all authors, the corresponding author states that there is no conflict of interest.

## AUTHOR AFFILIATIONS

[1]CHU Bordeaux, Medical Intensive Care Unit, Bordeaux, France
[2]Univ Bordeaux, Centre de Recherche Cardio-Thoracique de Bordeaux, Inserm UMR 1045, Bordeaux, France
[3]CHU Bordeaux, CRCM Pédiatrique, Bordeaux, France
[4]Mycology-Parasitology Department, CHU Bordeaux, Bordeaux, France

## AUTHOR ORCIDs

Renaud Prevel  http://orcid.org/0000-0003-2942-7350

## FUNDING

| Funder | Grant(s) | Author(s) |
|---|---|---|
| Federation Girondine de Lutte contre les Maladies Respiratoires Chroniques | | Renaud Prével |
| CHU Bordeaux | MD/PhD program | Raphaël Enaud |
| CHU Bordeaux | MD/PhD program | Renaud Prével |

## AUTHOR CONTRIBUTIONS

Arthur Orieux, Conceptualization, Data curation, Investigation, Writing – original draft | Raphaël Enaud, Formal analysis, Funding acquisition, Software, Writing – original draft | Sébastien Imbert, Investigation, Project administration, Validation, Writing – review and editing | Philippe Boyer, Data curation, Investigation, Writing – review and editing | Erwan Begot, Data curation, Investigation, Writing – review and editing | Adrian Camino, Data curation, Formal analysis, Investigation, Software, Writing – review and editing | Alexandre Boyer, Conceptualization, Project administration, Resources, Supervision, Validation, Visualization, Writing – review and editing | Patrick Berger, Methodology, Project administration, Resources, Supervision, Writing – review and editing | Didier Gruson, Conceptualization, Funding acquisition, Methodology, Project administration, Resources, Supervision, Validation, Writing – review and editing | Laurence Delhaes, Methodology, Project administration, Resources, Supervision, Validation, Writing – review and editing | Renaud Prevel, Conceptualization, Data curation, Formal analysis, Funding acquisition, Methodology, Project administration, Validation, Writing – original draft, Writing – review and editing

## DATA AVAILABILITY

The 16S rRNA gene and ITS2 sequences have been submitted to the European Nucleotide Archive (accession no. ERP134949). The scripts used for bioinformatics analysis during the current study are available in the Supplemental Materials.

## ETHICS APPROVAL

The study was approved by the Research Ethics Committee of Bordeaux University Hospital (reference CERBDX-2022-08) and performed according to The Code of Ethics of the World Medical Association (Declaration of Helsinki). According to French law and the French Data Protection Authority, the handling of these data for research purposes was declared to the Data Protection Officer of the Bordeaux University Hospital. Patients (or their relatives, if any) were notified about the anonymized use of their healthcare data via the department's booklet. In view of the documents at its disposal, the Research Ethics Committee of Bordeaux issued a favorable opinion for the publication of this research (Reference CERBDX-2022-08). An informed consent was obtained from all the patients or from their legal representatives.

## ADDITIONAL FILES

The following material is available online.

### Supplemental Material

**Supplemental Materials (Spectrum00641-23-s0001.docx).** Supplemental tables, figures, and script.

### Open Peer Review

**PEER REVIEW HISTORY (review-history.pdf).** An accounting of the reviewer comments and feedback.

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
