## [Reviewer comments · Microbiology Spectrum]

Microbiology Spectrum

The gut microbiota composition is linked to subsequent occurrence of ventilator-associated pneumonia in critically ill patients

Arthur Orieux, Raphaël Enaud, Sébastien Imbert, Philippe Boyer, Erwan Begot, Adrian Camino, Alexandre Boyer, Patrick Berger, Didier Gruson, Laurence Delhaes, and Renaud Prével

Corresponding Author(s): Renaud Prével, CHU Bordeaux

Review Timeline:

Submission Date:	February 12, 2023
Editorial Decision:	May 5, 2023
Revision Received:	July 6, 2023
Editorial Decision:	July 25, 2023
Revision Received:	July 26, 2023
Accepted:	July 26, 2023

Editor: Wei-Hua Chen

Reviewer(s): Disclosure of reviewer identity is with reference to reviewer comments included in decision letter(s). The following individuals involved in review of your submission have agreed to reveal their identity: Wei-Kai Wu (Reviewer #1); Chuanxin Wang (Reviewer #2)

Transaction Report:

DOI: <https://doi.org/10.1128/spectrum.00641-23>

May 5, 2023

Dr. Renaud Prével
Centre Hospitalier Universitaire de Bordeaux
bordeaux
France

Re: Spectrum00641-23 (The gut microbiota composition is linked to subsequent occurrence of ventilator-associated pneumonia in critically ill patients)

Dear Dr. Renaud Prével:

Thank you for submitting your manuscript to Microbiology Spectrum. Your manuscript has been evaluated by four external experts. Although they found your study interesting, they all raised concerns, some of which were critical. I thus would like to ask to address their concerns and revise your manuscript (and study) accordingly.

Link Not Available

Sincerely,

Wei-Hua Chen

Journals Department
Reviewer comments:

Reviewer #1 (Comments for the Author):

Orioux et al. conducted an observational study to investigate potential differences in rectal swab microbiota between critically intubated patients who developed ventilator-associated pneumonia (VAP) and those who did not. Of the 42 critically ill patients receiving invasive mechanical ventilation, 18 patients (43%) developed VAP, while 24 patients (57%) did not. The authors used 16S rRNA V3-V4 and ITS2 amplicon sequencing to analyze gut bacterial and fungal communities in the enrolled patients. They found no difference in alpha diversity (both bacterial and fungal) between patients with VAP and those without VAP. Regarding beta diversity, the authors observed borderline significant differences in gut mycobiota between patients with VAP and those without VAP ($p=0.05$). However, no significant difference in the beta diversity of gut bacteriota was noted ($p=0.56$). Additionally,

the authors identified some differential taxa between patients with VAP and those without VAP, including three bacterial species (*Bifidobacterium breve*, *Peptostreptococcus anaerobius*, and *Enterococcus avium*), a fungal genus (*Alternaria*), and fungal species (*Saccharomyces kudriavzevii*), which were enriched in patients who subsequently developed VAP, while *Megasphaera massiliensis* was enriched in patients who did not develop VAP. Generally, the clinical study design and microbiome analysis are technically sound; however, some issues may need to be addressed.

1. Since the study demonstrated no significant difference in the gut bacteriota profile (beta-diversity), emphasizing the increased presence of *Megasphaera massiliensis* in patients who did not develop VAP in the abstract may not be rational. I suggest the authors emphasize the enrichment of fungal taxa in patients who develop VAP, including the family of Pleosporaceae, genus of *Alternaria*, and *Saccharomyces kudriavzevii* because analysis of beta diversity revealed (despite borderline) significant differences in gut mycobiota composition, especially when the LDA scores of the fungal taxa are greater than bacterial taxa in LEfSe.
2. On page 1, line 31, "Patients with subsequent VAP had similar gut bacteriobiota and mycobiota α -diversities than those who did not develop VAP but gut mycobiota was dissimilar between these 2 groups." This sentence is difficult to understand. Please revise as "Patients who later developed VAP had similar gut bacteriobiota and mycobiota α -diversities compared to those who did not develop VAP. However, gut mycobiota composition was dissimilar between these two groups." I suggest the authors send the article for professional English editing to prevent misunderstandings thoroughly.
3. For Figure 2D and Figure 3D, I suggest the authors add a centroid with lines to the dots in each group to make the plot more easily visualized, especially when the beta diversity difference in Figure 3D is borderline.
4. Due to concerns about false positive findings frequently revealed by LEfSe, I suggest adding boxplots to statistically compare the abundances of the microbial taxa revealed in Figure 4A (4 bacterial species) and 4B (1 fungal family, 1 fungal genus, and 1 fungal species) between patients with VAP and those without VAP.
5. On page 22, line 557: "LefSe" should be corrected as "LEfSe."

Reviewer #2 (Comments for the Author):

In this article, the authors compared the gut microbiota composition in ventilator associated pneumonia in critically ill patients. The composition of the gut microbiota, but not α -diversity, differs between critically ill patients who subsequently develop VAP and those who do not. However, such manuscript probably not reach to the standards of our high quality journal now. My opinion is major revision.

The major concerns:

1. The author needs to check punctuation throughout, such as line 124.
2. The author's interpretation of the sequencing results is too shallow and not detailed enough, and should not be limited to diversity and population composition. For example, *Eur Respir J.* 2023 Feb 9 for details; 61 (2) : 2200910.
3. The author focused on the composition and differences of gut microbes, but as the author explained about the lung-gut axis, did the author focus on the pulmonary microbiome? I expect to see more discussion of this aspect in the results and discussion. For example, *Intensive Care Med.* 2021 Mar; 47 (3) : 292-306.
4. The author should update the references and cite more literatures from the last three years.
5. Whether the authors analyzed the ICU stay.
6. I wonder whether authors did subgroup analysis depending on probiotics.
7. The samples collected by the author were in the period of the epidemic of the 2019-nCoV. Did the author analyze this group separately?

Reviewer #3 (Comments for the Author):

Orioux et al. compare the bacterial and fungal components of the rectal microbiota from patients who had been on ventilators in the ICU for several days and either did or did not acquire ventilator associated pneumonia. They found that while the overall taxonomical alpha diversity did not differ between the two patient groups, there were some differences between the groups, primarily in the fungal communities.

Notes:

- 1) I think the bacteriobiota and mycobiota comparison figures should be moved from the supplement to the main text.
- 2) In Fig. S2 it looks like some of the samples (e.g. B193, B93, 10B, etc.) are not very different than the TN negative controls. Is there an explanation for this? Does this point to some of these results likely being due to noise/contamination?
- 3) Supplemental Table 1: Were those sample isolates present or absent in non VAP patients? If so, was their prevalence similar?
- 4) Fig. 3D. The differences between the groups are hard to visualize here. It looks like samples from the two groups almost completely overlap with one another, but the p-value is reported as 0.05. Is there a clearer way to present this data so that the differences can be seen more easily? I think something like a PCA plot with centroids would help significantly.

Reviewer #4 (Comments for the Author):

In their manuscript, Orieux et al. describe a 16S rRNA and ITS fecal metagenomic analysis of a cohort of patients from an intensive care unit to determine the contribution of the gut microbiota composition to ventilator associated pneumonia. They find that there are composition differences within the mycobiome (but not within bacterial compartment) of patients that go onto develop VAP compared to those who do not. Overall, while the manuscript describes a potentially interesting cohort, their analysis does not go beyond a superficial description of the cohort and does not provide any new insights into how gut microbes would influence the development of VAP.

Major comments:

- There is very little explanation given for any of the analyses presented in their manuscript. The rationale and hypothesis underlying their study is not presented in the introduction. Likewise, throughout the results, figures are referred to with very little or no explanation of their analysis and conclusions.
- The analysis of their data was superficial. For example, the only real difference observed between VAP and controls was within the mycobiome (by ITS sequencing). Yet, there was no explanation about how difference in fungal gut composition would be related to a bacterial pneumonia. There were many missed opportunities for informative analyses that do not appear to have been performed--were the VAP causative organisms present in the gut microbiota?
- Comparisons of compositional data did not take into account demographic/confounding factors that could shape gut microbiome composition, which greatly weakens the finding of compositional differences between VAP and controls. PERMANOVA can be used to estimate the contributions of confounders, but this does not appear to have been performed.
- There was no description of controls used to mitigate batch effect or differences sequence depth.

Minor comments:

- Line 131-132, it is unclear what "average read counts" represent. Reads/ASVs per sample? Per patient?
- Methods: Sup fig 1 and 2 are not especially helpful. More useful would be to report the number of reads and/or the taxa found in negative controls and explain if they were observed in samples, and if so, how they were dealt with.
- Line 31: Confusing sentence. Why is beta-diversity not mentioned.
- Line 59-60: "% days" doesn't make sense.
- Sup 3 and 4 require more explanation. Text describing this was confusing. Was this finding expected? How were differences between beta diversities determined to be significant? Presumably PERMANOVA but this is not stated.
- Sup 5 and 6. Statistical test for these figures not described.

Staff Comments:

Preparing Revision Guidelines

Please return the manuscript within 60 days; if you cannot complete the modification within this time period, please contact me. If you do not wish to modify the manuscript and prefer to submit it to another journal, please notify me of your decision immediately so that the manuscript may be formally withdrawn from consideration by Microbiology Spectrum.

Comments to the Author

In this article, the authors compared the gut microbiota composition in ventilator associated pneumonia in critically ill patients. The composition of the gut microbiota, but not α -diversity, differs between critically ill patients who subsequently develop VAP and those who do not. However, such manuscript probably not reach to the standards of our high quality journal now. My opinion is major revision.

The major concerns:

1. The author needs to check punctuation throughout, such as line 124.
2. The author's interpretation of the sequencing results is too shallow and not detailed enough, and should not be limited to diversity and population composition. For example, *Eur Respir J.* 2023 Feb 9 for details; 61 (2) : 2200910.
3. The author focused on the composition and differences of gut microbes, but as the author explained about the lung-gut axis, did the author focus on the pulmonary microbiome? I expect to see more discussion of this aspect in the results and discussion. For example, *Intensive Care Med.* 2021 Mar; 47 (3) : 292-306.
4. The author should update the references and cite more literatures from the last three years.
5. Whether the authors analyzed the ICU stay.
6. I wonder whether authors did subgroup analysis depending on probiotics.

7. The samples collected by the author were in the period of the epidemic of the 2019-nCoV. Did the author analyze this group separately?

Dear Editor,

Thank you for assessing our work for potential publication in Microbiology Spectrum and for having it peer-reviewed.

We also would like to thank both Reviewers for their relevant comments which helped us to improve this manuscript.

Please find thereafter the responses we address to Reviewer's comments.

Reviewer #1 (Comments for the Author):

Orieux et al. conducted an observational study to investigate potential differences in rectal swab microbiota between critically intubated patients who developed ventilator-associated pneumonia (VAP) and those who did not. Of the 42 critically ill patients receiving invasive mechanical ventilation, 18 patients (43%) developed VAP, while 24 patients (57%) did not. The authors used 16S rRNA V3-V4 and ITS2 amplicon sequencing to analyze gut bacterial and fungal communities in the enrolled patients. They found no difference in alpha diversity (both bacterial and fungal) between patients with VAP and those without VAP. Regarding beta diversity, the authors observed borderline significant differences in gut mycobiota between patients with VAP and those without VAP ($p=0.05$). However, no significant difference in the beta diversity of gut bacteriota was noted ($p=0.56$). Additionally, the authors identified some differential taxa between patients with VAP and those without VAP, including three bacterial species (*Bifidobacterium breve*, *Peptostreptococcus anaerobius*, and *Enterococcus avium*), a fungal genus (*Alternaria*), and fungal species (*Saccharomyces kudriavzevii*), which were enriched in patients who subsequently developed VAP, while *Megasphaera massiliensis* was enriched in patients who did not develop VAP. Generally, the clinical study design and microbiome analysis are technically sound; however, some issues may need to be addressed.

1. Since the study demonstrated no significant difference in the gut bacteriota profile (beta-diversity), emphasizing the increased presence of *Megasphaera massiliensis* in patients who did not develop VAP in the abstract may not be rational. I suggest the authors emphasize the enrichment of fungal taxa in patients who develop VAP, including the family of Pleosporaceae, genus of *Alternaria*, and *Saccharomyces kudriavzevii* because analysis of beta diversity revealed (despite borderline) significant differences in gut mycobiota composition, especially when the LDA scores of the fungal taxa are greater than bacterial taxa in LEfSe.

We emphasized *Megasphaera massiliensis* as it was associated with the absence of subsequent VAP and because it was the only microbial specie also found to be associated with the occurrence of VAP by ANCOM-BC analysis (see response to Comment 4).

Nevertheless, we do agree that the association of lung fungi with the occurrence of bacterial VAP deserves to be further discussed as the association of the genus *Alternaria sp.* with the occurrence of VAP was also confirmed by ANCOM-BC analysis.

We so added in the abstract:

"whereas the presence of the fungal genus *Alternaria sp.* was associated with the occurrence of VAP."

We also added in the discussion section:

"Notably, differential abundance testing methods often provide different results and it is now recommended to confront at least two differential abundance methods to help ensure robust biological interpretations (59). Using this multiple tests method, the fungal genus *Alternaria sp.* was found to be associated with the occurrence of subsequent VAP. *Alternaria sp.* is known to increase lung inflammation in murine models of asthma via tuft cell-produced cysteinyl leukotrienes and IL-25 (60, 61)."

2. On page 1, line 31, "Patients with subsequent VAP had similar gut bacteriobiota and mycobiota α -diversities than those who did not develop VAP but gut mycobiota was dissimilar between these 2 groups." This sentence is difficult to understand. Please revise as "Patients who later developed VAP had similar gut bacteriobiota and mycobiota α -diversities compared to those who did not develop VAP. However, gut mycobiota composition was dissimilar between these two groups." I suggest the authors send the article for professional English editing to prevent misunderstandings thoroughly.

We corrected the sentence as requested. English editing was performed by Dr Shradha Wali who is an English native speaker.

3. For Figure 2D and Figure 3D, I suggest the authors add a centroid with lines to the dots in each group to make the plot more easily visualized, especially when the beta diversity difference in Figure 3D is borderline.

Figure 2D and 3D (now 3D and 4D, respectively) and Figures S3D and S4D were modified as requested.

4. Due to concerns about false positive findings frequently revealed by LEfSe, I suggest adding boxplots to statistically compare the abundances of the microbial taxa revealed in Figure 4A (4 bacterial species) and 4B (1 fungal family, 1 fungal genus, and 1 fungal species) between patients with VAP and those without VAP.

We do agree with Reviewer 1 about the high variability of the results provided by differential abundance testing methods and that's the reason why we emphasized in the discussion section only the bacterial specie confirmed by a second approach (ANCOM-BC) (see Comment 1).

We so provided further precision about this confirmatory step by ANCOM-BC method in the results section: "Using LDA analysis to detect significant microbial community

enrichment, we identified *Megasphaera massiliensis* as being associated with the absence of VAP occurrence (LDA score > 3) (Figure 4A), association confirmed by ANCOM-BC analysis ($p < 0.001$). On the contrary, several bacterial species (*Bifidobacterium breve*, *Peptostreptococcus anaerobius* and *Enterococcus avium*) but this association was not confirmed by ANCOM-BC analysis. This association was driven by the fact that *M. massiliensis* was not observed in any gut microbiota of patients who will develop subsequent VAP (Supplemental Figure 5A). Regarding fungi, the fungal genus (*Alternaria sp.*) and fungal species (*Saccharomyces kudriavzevii*) were associated with subsequent VAP occurrence (Figure 4B), but this association was only confirmed for *Alternaria sp.* by ANCOM-BC analysis ($p < 0.0001$). *Alternaria sp.* was only found in the gut microbiota of patients who will develop subsequent VAP (Supplemental Figure 5B)."

We also provide the reads abundance in Supplemental Figure 5A and 5B.

We also underlined this point in the Discussion section: "Notably, differential abundance testing methods often provide different results and it is now recommended to confront at least two differential abundance methods to help ensure robust biological interpretations (59). Using this multiple tests method, (...)

5. On page 22, line 557: "LefSe" should be corrected as "LEfSe."

This typing error is corrected.

Reviewer #2 (Comments for the Author):

In this article, the authors compared the gut microbiota composition in ventilator associated pneumonia in critically ill patients. The composition of the gut microbiota, but not α -diversity, differs between critically ill patients who subsequently develop VAP and those who do not. However, such manuscript probably not reach to the standards of our high quality journal now. My opinion is major revision.

The major concerns:

1. The author needs to check punctuation throughout, such as line 124.

We checked punctuation as requested and English editing was performed by Dr Shradha Wali who is an English native speaker.

2. The author's interpretation of the sequencing results is too shallow and not detailed enough, and should not be limited to diversity and population composition. For example, Eur Respir J. 2023 Feb 9 for details; 61 (2) : 2200910.

We read this interesting paper with interest when it was published and we thank Reviewer 2 for providing it.

In this paper, the authors assessed the bacterial density (log₁₆S copies per sample) after droplet digital PCR which was not performed in our analysis. After short-read (or targeted) NGS sequencing, the recommended expression of sample diversity is based on α -diversity indices such as Shannon, Simpson or evenness as we provide in our manuscript (F. Finotello, E. Mastorilli, B. Di Camillo Measuring the diversity of the human microbiota with targeted next-generation sequencing. Brief Bioinform, 19 (2016), pp. 679-692). As the results from different α -diversity indices can vary, we provide the results from three different ones to assess consistency (K.V.-A. Johnson, P.W.J. Burnet Microbiome: should we diversify from diversity? Gut Microbes, 7 (2016), pp. 455-458). But we can't assess proper bacterial load from our data.

As we found no difference in alpha diversity (both bacterial and fungal) between patients with VAP and those without VAP, univariate logistic regression between alpha diversity indices and occurrence of VAP was not relevant.

To provide more insight into phyla relative abundance in the main manuscript, we moved the figures from supplementary materials to the main manuscript.

We also detailed more if the causative VAP bacterial species were already present in the gut microbiota at admission in the results section:

"Of note, the causative VAP bacteria was present in the gut microbiota in 9/18 patients (50%) who will develop subsequent VAP with a median proportion of reads related to the causative bacteria compared to the numbers of reads after filtration for the corresponding sample of 0,66%, IQR [0,39-0,95]. The species involved were *K. pneumoniae* (n=3), *E. coli* (n=2), *K. variicola* (n=1), *Acinetobacter baumannii* (n=1),

Enterococcus faecalis (n=1), *Enterococcus faecium* (n=1). The median time between orotracheal intubation (OTI) and VAP occurrence was 6 days, IQR [3-8] with a trend for a shorter time in patients with the causative bacteria detected in the gut microbiota at admission compared to the others (median time 4 days, IQR [3-6] vs 8 days, IQR [4-17], $p=0.09$)."

3. The author focused on the composition and differences of gut microbes, but as the author explained about the lung-gut axis, did the author focus on the pulmonary microbiome? I expect to see more discussion of this aspect in the results and discussion. For example, *Intensive Care Med.* 2021 Mar; 47 (3) : 292-306.

We better justify why we focused on the gut microbiota in the introduction part:

This role can be mediated via microbial direct interactions alleviating or enhancing host colonization resistance and the emergence of pathogens (9, 10). Microbiota also interacts with the host modulating the local and systemic immune system (11, 12). Lung microbiota was logically the more extensively investigated compartment as VAP develop within the lungs (13). Reduced lung bacteriobiota α -diversity was associated with increased pathogenic bacterial presence and increased lung inflammation (14). Consistent with the concept of transcolonization, the composition of oropharyngeal and gut microbiota could influence lung microbiota composition (8, 15). The composition of gut microbiota can also enhance host resistance to lung bacterial infection via lung immune system modulation (12, 16). Despite this pre-clinical data, the role of gut microbiota in VAP occurrence has never been investigated.

We also added in the discussion section:

Because of the more extensive data regarding oropharyngeal and lung microbiota but the absence of data regarding gut microbiota, we decided to explore, for the first time, the role of gut microbiota in VAP occurrence.

Nevertheless, the lack of concomitant lung and oropharyngeal samples is a first limitation of our study, as lung microbiota is dependent on gut and oropharyngeal microbiota as described in the "gut-lung axis" and "transcolonization" concepts. It would be relevant to investigate the interplay between the microbiota of these three compartments and its impact on the development of VAP longitudinally.

4. The author should update the references and cite more literatures from the last three years.

We added the following citations:

2020:

Enaud R, Prevel R, Ciarlo E, Beaufils F, Wieërs G, Guery B, Delhaes L. 2020. The gut-lung axis in health and respiratory diseases: a place for inter-organ and inter-kingdom crosstalks. *Front Cell Infect Microbiol* 10:9.

2021:

Fromentin M, Ricard J-D, Roux D. 2021. Respiratory microbiome in mechanically ventilated patients: a narrative review. *Intensive Care Med* 47:292–306.

Fenn D, Abdel-Aziz MI, van Oort PMP, Brinkman P, Ahmed WM, Felton T, Artigas A, Póvoa P, Martin-Loeches I, Schultz MJ, Dark P, Fowler SJ, Bos LDJ, BreathDx Consortium. 2022. Composition and diversity analysis of the lung microbiome in patients with suspected ventilator-associated pneumonia. *Crit Care* 26:203.

Durán-Manuel EM, Loyola-Cruz MÁ, Cruz-Cruz C, Ibáñez-Cervantes G, Gaytán-Cervantes J, González-Torres C, Quiroga-Vargas E, Calzada-Mendoza CC, Cureño-Díaz MA, Fernández-Sánchez V, Castro-Escarpulli G, Bello-López JM. 2022. Massive sequencing of the V3-V4 hypervariable region of bronchoalveolar lavage from patients with COVID-19 and VAP reveals the collapse of the pulmonary microbiota. *J Med Microbiol* 71.

2023:

Gregorczyk-Maga I, Pałka A, Fiema M, Kania M, Kujawska A, Maga P, Jachowicz-Matczak E, Romaniszyn D, Chmielarczyk A, Żółtowska B, Wójkowska-Mach J. 2023. Impact of tooth brushing on oral bacteriota and health care-associated infections among ventilated COVID-19 patients: an intervention study. *Antimicrobial Resistance & Infection Control* 12:17.

Cotoia A, Paradiso R, Ferrara G, Borriello G, Santoro F, Spina I, Mirabella L, Mariano K, Fusco G, Cinnella G, Singer P. 2023. Modifications of lung microbiota structure in traumatic brain injury ventilated patients according to time and enteral feeding formulas: a prospective randomized study. *Crit Care* 27:244.

Chen Y-H, Yeung F, Lacey KA, Zalana K, Lin J-D, Bee GCW, McCauley C, Barre RS, Liang S-H, Hansen CB, Downie AE, Tio K, Weiser JN, Torres VJ, Bennett RJ, Loke P, Graham AL, Cadwell K. 2023. Rewilding of laboratory mice enhances granulopoiesis and immunity through intestinal fungal colonization. *Science Immunology* 8:eadd6910.

Schlechte J, Zucoloto AZ, Yu I-L, Doig CJ, Dunbar MJ, McCoy KD, McDonald B. 2023. Dysbiosis of a microbiota-immune metasytem in critical illness is associated with nosocomial infections. *Nat Med*.

5. Whether the authors analyzed the ICU stay.

We analysed the ICU stay before the occurrence of VAP as stated in the results section

The median time between orotracheal intubation (OTI) and VAP occurrence was 6 days, IQR [3-8].

But we did not analyse the length of ICU stay, as it has already been demonstrated that VAP increases this length.

6. I wonder whether authors did subgroup analysis depending on probiotics.

It is not a routine procedure in our ICU so we stated in the first results paragraph:

None of the patients received prebiotics, probiotics or selective digestive decontamination.

7.The samples collected by the author were in the period of the epidemic of the 2019-nCoV. Did the author analyze this group separately?

The onset of the 1st wave occurred in March 2020 in our city and we stopped the inclusion at this date. No COVID-19 patient was included in this study. We added in the Methods section: "from October 2019 to March 2020 (stopped at COVID-19 wave occurrence)."

Reviewer #3 (Comments for the Author):

Orieux et al. compare the bacterial and fungal components of the rectal microbiota from patients who had been on ventilators in the ICU for several days and either did or did not acquire ventilator associated pneumonia. They found that while the overall taxonomical alpha diversity did not differ between the two patient groups, there were some differences between the groups, primarily in the fungal communities.

Notes:

1) I think the bacteriobiota and mycobiota comparison figures should be moved from the supplement to the main text.

We moved these figures to the main text as requested.

2) In Fig. S2 it looks like some of the samples (e.g. B193, B93, 10B, etc.) are not very different than the TN negative controls. Is there an explanation for this? Does this point to some of these results likely being due to noise/contamination?

We do agree with Reviewer 3 that some of these samples are pretty close to TN negative controls. The explanation for this is that lower airway microbiota is a low-biomass environment, even more regarding the mycobiota as there are 4 to 10 times less fungi than bacteria within the lungs. Whether include or exclude these samples is a relevant question with the underlying concern about the quality of the sample. As 16S and ITS amplification PCR were performed on the DNA from the same sample and that FigS1 suggests that the samples are different from negative controls, we assumed that our samples were of sufficient quality to be included in the analysis and that the only light difference observed in FigS2 is due to the low fungal biomass and not to a flawed sampling.

3) Supplemental Table 1: Were those sample isolates present or absent in non VAP patients? If so, was their prevalence similar?

These isolates were present only in VAP patients. We changed the first section to make it clearer: "Bacterial species isolated on pulmonary samples from patients with ventilator-associated pneumonia"

4) Fig. 3D. The differences between the groups are hard to visualize here. It looks like samples from the two groups almost completely overlap with one another, but the p-value is reported as 0.05. Is there a clearer way to present this data so that the differences can be seen more easily? I think something like a PCA plot with centroids would help significantly.

Figure 2D and 3D (now 3D and 4D, respectively) and Figures S3D and S4D were modified as requested.

Reviewer #4 (Comments for the Author):

In their manuscript, Orioux et al. describe a 16S rRNA and ITS fecal metagenomic analysis of a cohort of patients from an intensive care unit to determine the contribution of the gut microbiota composition to ventilator associated pneumonia. They find that there are composition differences within the mycobiome (but not within bacterial compartment) of patients that go onto develop VAP compared to those who do not. Overall, while the manuscript describes a potentially interesting cohort, their analysis does not go beyond a superficial description of the cohort and does not provide any new insights into how gut microbes would influence the development of VAP.

Major comments:

- There is very little explanation given for any of the analyses presented in their manuscript. The rationale and hypothesis underlying their study is not presented in the introduction. Likewise, throughout the results, figures are referred to with very little or no explanation of their analysis and conclusions.

We developed in the introduction the rationale to investigate the role of gut microbiota in the occurrence of VAP:

“This role can be mediated via microbial direct interactions alleviating or enhancing host colonization resistance and emergence of pathogens (9, 10). Microbiota also interacts with the host modulating the local and systemic immune system (11, 12). Lung microbiota was logically the compartment the more extensively investigated as VAP develop within the lungs (13). Reduced lung bacteriobiota α -diversity was associated with increased pathogenic bacterial presence and increased lung inflammation (14). Consistent with the concept of transcolonization, the composition of oropharyngeal and gut microbiota could influence lung microbiota composition (8, 15). The composition of gut microbiota can also enhance host resistance to lung bacterial infection via lung immune system modulation (12, 16). Despite this pre-clinical data, the role of gut microbiota in VAP occurrence has never been investigated.”

- The analysis of their data was superficial. For example, the only real difference observed between VAP and controls was within the mycobiome (by ITS sequencing). Yet, there was no explanation about how difference in fungal gut composition would be related to a bacterial pneumonia. There were many missed opportunities for informative analyses that do not appear to have been performed--were the VAP causative organisms present in the gut microbiota?

To better explain how difference in fungal gut composition can be related to a bacterial pneumonia, we extended the related discussion section:

“Even if it can be disturbing to imagine a role of gut microbiota in the development of a long-distance, other-kingdom infection such as bacterial VAP, the role of gut mycobiota in VAP should not be underestimated as demonstrated by the dissimilarity in gut

mycobiota composition between patients who develop VAP or not observed in this study. In fact, direct inter-kingdom interactions occur within the microbiota (8) and could modulate the dynamics of the bacteriobiota and/or the expansion of any pathogen. For instance, *E. coli* and *Candida albicans* have been demonstrated to co-evolve within biofilms conferring increased ofloxacin tolerance to *E. coli* (28). Besides, commensal fungi are major drivers of gut barrier integrity (29) and could help to limit bacterial translocation from the gut. Moreover, commensal fungi also have an indirect impact through the modulation of local and systemic immunity (30) enhancing the plausible relevant impact of gut mycobiota in subsequent development of VAP. This point has recently been highlighted by the enhanced granulopoiesis after rewilding laboratory mice via fungal colonizing, conferring long-term protection from bacterial infection to these mice (31). This study confirms a previous one which demonstrated that commensal fungi recapitulate the protective benefits of intestinal bacteria (30). Interestingly, a link between decreased gut microbiota diversity and increased rate of nosocomial infections via alteration of neutrophils functions has recently been suggested but investigation of mycobiota was lacking (32). Consequently, the fungal kingdom of microbiota should be investigated in future studies."

Regarding the VAP causative organisms, we performed the analysis required by Reviewer 4 and completed in the Results section:

"Of note, the causative VAP bacteria was present in the gut microbiota in 9/18 patients (50%) who will develop subsequent VAP with a median proportion of reads related to the causative bacteria compared to the numbers of reads after filtration for the corresponding sample of 0,66%, IQR [0,39-0,95]. The species involved were *K. pneumoniae* (n=3), *E. coli* (n=2), *K. variicola* (n=1), *Acinetobacter baumannii* (n=1), *Enterococcus faecalis* (n=1), *Enterococcus faecium* (n=1). The median time between orotracheal intubation (OTI) and VAP occurrence was 6 days, IQR [3-8] with a trend for a shorter time in patients with the causative bacteria detected in the gut microbiota at admission compared to the others (median time 4 days, IQR [3-6] vs 8 days, IQR [4-17], $p=0.09$)."

- Comparisons of compositional data did not take into account demographic/confounding factors that could shape gut microbiome composition, which greatly weakens the finding of compositional differences between VAP and controls. PERMANOVA can be used to estimate the contributions of confounders, but this does not appear to have been performed.

We do agree with the Reviewer that deciphering the factors shaping gut microbiota composition is a relevant question in this field but it is not the point of our study. In fact, the aim of our study is to assess a link between gut microbiota composition and the occurrence of VAP not matter what is the cause of this composition.

To answer the Reviewer's question, much larger cohorts are needed. In fact, gut microbiota composition is dependent on many factors in critically patients both endogenous (increased production of opioids, decreased bile-salt concentration, gastrointestinal dysmotility, increased production of catecholamines, loss of epithelial integrity in the intestine) and exogenous (antibiotics, SOD/SDD, gastric-acid inhibition, enteral/parenteral feeding, sedatives, opioids, catecholamines) (Haak BW, Lancet 2017). Adjustment on all these variables requires much larger cohorts, including hundreds of patients.

Nevertheless, we stated this point in our manuscript: "Even if it was not the research question of this study, adjustment on all these variables would require larger cohorts including hundreds of patients."

Finally, we do agree that our study does not provide causality but only association as stated in the Discussion section: "Finally, our study provides correlation but not causality demonstration".

- There was no description of controls used to mitigate batch effect or differences sequence depth.

Due to the number of samples, DNA extraction was performed with the same kit, amplification PCR on the same day and sequencing on the same flow cell, thus controlling batch effect and differences in sequence depth.

Library sizes were obtained as follow:

16S

ITS2

Rarefaction curves are presented thereafter:

16S

ITS2

Minor comments:

- Line 131-132, it is unclear what "average read counts" represent. Reads/ASVs per sample? Per patient?

We added in the Methods section: "The final average read counts (amplicons after quality filtration but before assignment to obtain ASVs) per sample were (...)"

- Methods: Sup fig 1 and 2 are not especially helpful. More useful would be to report the number of reads and/or the taxa found in negative controls and explain if they were observed in samples, and if so, how they were dealt with.

Phyla with only one taxa and ASV in only 3 or less samples were removed from the analysis to avoid false sequences due to sequencing errors.

We performed analysis without and with a decontamination step (MicroDecon) finding no difference.

- Line 31: Confusing sentence. Why is beta-diversity not mentioned.

We added β -diversity "However, gut mycobiota was dissimilar (β -diversity) between these 2 groups" to clarify this point.

- Line 59-60: "% days" doesn't make sense.

We modified the sentence to: (about 10 to 25 VAP for 1,000 days of mechanical ventilation)

- Sup 3 and 4 require more explanation. Text describing this was confusing. Was this finding expected? How were differences between beta diversities determined to be significant? Presumably PERMANOVA but this is not stated.

The statistic test, i.e. Bray-Curtis dissimilarity, is stated both in the Methods section and the Supplemental Figure legends and we added the word PERMANOVA in the Supplemental figure legends:

"Between sample β -diversity differences (measured using Bray Curtis dissimilarity) were tested using a permutational multivariate ANOVA (PERMANOVA)"

We also added PERMANOVA in the Figure legends.

Supplemental Figure 3. Comparison of gut bacteriobiota between patients who received antimicrobial therapy within the past 3 months of admission and those who did not. A. Boxplot of estimated α -diversity by Shannon index. B. Boxplot of estimated α -diversity by Simpson index. C. Boxplot of estimated α -diversity by evenness. D. Non-metric Bray-

curtis analysis of β -diversity (PERMANOVA). Threshold for statistical significance: $p=0.05$. atb: antibiotics.

Supplemental Figure 4. Comparison of gut mycobiota between patients who received antimicrobial therapy within the past 3 months of admission and those who did not. A. Boxplot of estimated α -diversity by Shannon index. B. Boxplot of estimated α -diversity by Simpson index. C. Boxplot of estimated α -diversity by evenness. D. Non-metric Bray-curtis analysis of β -diversity (PERMANOVA). Threshold for statistical significance: $p=0.05$. atb: antibiotics.

The absence of difference regarding gut bacteriobiota α -diversity between patients who received antimicrobial therapy within the past 3 months before admission and those who did not is consistent with the resiliency of gut microbiota diversity within weeks after antibiotics administration already observed, even if this this resiliency is impaired after several antimicrobial courses (Dethlefsen *et al.*, PNAS 2011).

We stated:

“This absence of difference not is consistent with the resiliency of gut microbiota diversity within weeks after antibiotics administration already observed, even if this resiliency is impaired after several antimicrobial courses (56).”

- Sup 5 and 6. Statistical test for these figures not described.

The statistical test is now clarified in both the Figure legend and the Methods section (the Sup Figures were moved to the main manuscript according to Reviewer 3 request.

Figure 2. Comparison of gut microbiota phylum relative abundance between patients who subsequently develop VAP and those who do not. VAP: patients who subsequently develop VAP. no_VAP: patients who do not. Wilcoxon rank sum test. Threshold for statistical significance: $p=0.05$

For cross-sectional analyses, at a specific time, significant differences in phyla abundance and in α -diversity were determined using the Mann-Whitney Wilcoxon rank-sum test.

We hope that these precisions will make you consider the possibility to submit a revised manuscript for potential publication in Microbiology Spectrum.

Very respectfully,

Drs Arthur Orioux, Raphaël Enaud and Renaud Prével

July 25, 2023

Dr. Renaud Prével
CHU Bordeaux
Place Amélie Raba Léon
Bordeaux
France

Re: Spectrum00641-23R1 (The gut microbiota composition is linked to subsequent occurrence of ventilator-associated pneumonia in critically ill patients)

Dear Dr. Renaud Prével:

Thank you for submitting your manuscript to Microbiology Spectrum. As you will see your paper is very close to acceptance. Please modify the manuscript along the lines I have recommended. As these revisions are quite minor, I expect that you should be able to turn in the revised paper in less than 30 days, if not sooner. You will find the reviewers' comments below.

When submitting the revised version of your paper, please provide (1) point-by-point responses to the issues raised by the reviewers as file type "Response to Reviewers," not in your cover letter, and (2) a PDF file that indicates the changes from the original submission (by highlighting or underlining the changes) as file type "Marked Up Manuscript - For Review Only". Please use this link to submit your revised manuscript. Detailed instructions on submitting your revised paper are below.

Link Not Available

Sincerely,

Wei-Hua Chen

Reviewer comments:

Reviewer #1 (Comments for the Author):

The authors emphasized that they used ANCOM-BC to confirm their findings of the differential taxonomies, however, I do not see any analytical result conducted by ANCOM-BC in the manuscript or figures. The authors should reveal the result of significant differential taxa identified by ANCOM-BC, not just showing $p < 0.0001$ for a specific taxa in the text (i.e. *Megasphaera massiliensis*).

Preparing Revision Guidelines

- Point-by-point responses to the issues raised by the reviewers in a file named "Response to Reviewers," NOT IN YOUR COVER LETTER.
- Upload a compare copy of the manuscript (without figures) as a "Marked-Up Manuscript" file.
- Each figure must be uploaded as a separate file, and any multipanel figures must be assembled into one file.

- Manuscript: A .DOC version of the revised manuscript
- Figures: Editable, high-resolution, individual figure files are required at revision, TIFF or EPS files are preferred

Please return the manuscript within 60 days; if you cannot complete the modification within this time period, please contact me. If you do not wish to modify the manuscript and prefer to submit it to another journal, please notify me of your decision immediately so that the manuscript may be formally withdrawn from consideration by Microbiology Spectrum.

Dear Editor,

Thank you for assessing our work for potential publication in *Microbiology Spectrum* and for having it peer-reviewed.

Please find thereafter the response we address to Reviewer's comments.

Reviewer #1 (Comments for the Author):

The authors emphasized that they used ANCOM-BC to confirm their findings of the differential taxonomies, however, I do not see any analytical result conducted by ANCOM-BC in the manuscript or figures. The authors should reveal the result of significant differential taxa identified by ANCOM-BC, not just showing $p < 0.0001$ for a specific taxa in the text (i.e. *Megasphaera massiliensis*).

We provide raw data from ANCOM-BC analysis in the supplementary materials, assessing it in the results section of the manuscript :

"The results of ANCOM-BC analysis are provided in Supplemental Table 2 and 3 for bacteriobiota and mycobiota analysis, respectively."

We hope that these precisions will make you consider the possibility to submit a revised manuscript for potential publication in *Microbiology Spectrum*.

Very respectfully,

Drs Arthur Orioux, Raphaël Enaud and Renaud Prével

July 26, 2023

Dr. Renaud Prével
CHU Bordeaux
Place Amélie Raba Léon
Bordeaux
France

Re: Spectrum00641-23R2 (The gut microbiota composition is linked to subsequent occurrence of ventilator-associated pneumonia in critically ill patients)

Dear Dr. Renaud Prével:

Your manuscript has been accepted, and I am forwarding it to the ASM Journals Department for publication. You will be notified when your proofs are ready to be viewed.

Sincerely,

Wei-Hua Chen
Editor, Microbiology Spectrum
